# Fostering Empathy, Implicit Bias Mitigation, and Compassionate Behavior in a Medical Humanities Course

**DOI:** 10.3390/ijerph17072169

**Published:** 2020-03-25

**Authors:** Brian D. Schwartz, Alexis Horst, Jenifer A. Fisher, Nicole Michels, Lon J. Van Winkle

**Affiliations:** Department of Medical Humanities, Rocky Vista University, 8401 S. Chambers Road, Parker, CO 80134, USA; bschwartz@rvu.edu (B.D.S.); ahorst@rvu.edu (A.H.); jfisher@rvu.edu (J.A.F.); nmichels@rvu.edu (N.M.)

**Keywords:** compassion, empathy, observational study, reflective capacity, team-based learning, implicit bias

## Abstract

Increases in compassionate behavior improve patient outcomes and reduce burnout among healthcare professionals. We predicted that selecting and performing service-learning projects by teams of prospective medical students in a Medical Humanities course would foster students’ compassion by raising their reflective capacity, empathy, and unconscious bias mitigation. In class, we discussed difficulties in communication and implicit bias. In this observational study, teams wrote individual and team critical reflections on these class discussions and their service-learning experiences, and we analyzed these reflections for dissonance, self-examination, bias mitigation, dissonance reconciliation, and compassionate behavior. Thirty-two students (53% female) completed the Reflective Practice Questionnaire and the Jefferson Scale of Empathy before the course in August 2019 and after it in December 2019. In December, students were surveyed concerning their attitudes toward team service-learning projects and unconscious bias. The students reported changes in their behavior to mitigate biases and become more compassionate, and their reflective capacity and empathy grew in association with discussions and team service-learning experiences in the course. Virtually all students agreed with the statement “Unconscious bias might affect some of my clinical decisions or behaviors as a healthcare professional,” and they worked to control such biases in interactions with the people they were serving.

## 1. Introduction

Curricula to educate premedical, medical, and other healthcare professional students should include efforts to foster compassionate behavior, because such behavior likely causes healthcare providers as well as their patients to be happier [1,2]. Humanistic values are believed to “animate” compassionate professional behavior [3], and these values include accountability, altruism, duty, excellence, honor, integrity, and respect for others [4]. The application of these values is, however, difficult without concrete experiences on which to reflect [5]. Thus, we have attempted to foster more humane professional behavior in teams of premedical, medical, and healthcare professional students through the selection and performance of service-learning projects [6,7,8].

In a prior study, reported in this Special Issue, we found that direct and vivid experiences of selecting and performing service-learning projects did, indeed, foster dissonance, self-examination, bias mitigation, dissonance reconciliation, and compassionate behavior in teams of prospective medical students [8]. These projects fostered critical reflection to reconcile dissonance and mitigate bias by students. In critical reflection, one recognizes how their thoughts and behaviors do not match their personal and humanistic values, experiences perplexity, doubt, hesitation, or mental difficulties (i.e., dissonance), and decides how better to align their values, thoughts, and behaviors (i.e., dissonance reconciliation) [7,9,10].

Prospective medical students also expressed positive attitudes toward their teams, team-based learning, community service, and recognizing/mitigating their implicit biases [8]. Similarly, reflections on team service-learning experiences were accompanied by increases in the students’ reflective capacity (RC), as measured by a reliable survey of this characteristic [11,12]. Finally, students’ RC scores correlated positively with their cognitive empathy scores (a component of compassion) [8].

For these reasons, we investigated whether increases in students’ RC are associated with improved cognitive empathy scores. We also wondered whether an increased emphasis on understanding the nature of implicit bias in our Medical Humanities course, and the resultant pedagogy of discomfort [13,14], would cause students to accept the possibility that unconscious bias might affect some of their clinical decisions or behaviors as healthcare professionals. In our prior study, students reported bias mitigation in their written reflections on service learning, so that bias needed to become explicit rather than remain implicit [8]. However, a course to learn about implicit bias still left 22% of medical students able to deny that implicit bias might affect some of their clinical decisions and behaviors [15,16].

We also reported previously that medical and prospective healthcare professional students remain neutral as groups when asked whether performing service learning caused them to study for their courses with more interest than would have occurred without the project [6,7,8]. In the present study, we ask students to report the relationships between their service-learning projects and the basic science courses in which they were enrolled. As a result, we hoped they would report the impact of service learning on their interest in studying for all of their courses. Our specific hypotheses were as follows:

**Hypothesis** **1.**
*Our correct predictions for 52 previous participants (i.e., hypotheses a-d below [8]) will be confirmed for 34 new students.*

*The direct and vivid experiences of selecting and performing service-learning projects by teams of prospective medical students in a Medical Humanities course will be associated with written reflections exhibiting dissonance, self-examination, bias mitigation, dissonance reconciliation, and compassionate behavior.*

*Students will express positive attitudes toward their teams, team-based learning, community service, and recognizing/mitigating their implicit biases in association with team service-learning experiences.*

*Reflections on team service-learning experiences will be accompanied by increases in students’ reflective capacity, as measured by a reliable survey of this characteristic.*

*The students’ reflective capacity scores will correlate positively with their cognitive empathy scores (a component of compassion).*



**Hypothesis** **2.**
*Increases in students’ reflective capacity scores will be associated with increases in their cognitive empathy scores.*


**Hypothesis** **3.**
*A new emphasis on implicit bias in our course, through discussions of implicit association test results, will be associated with acceptance that unconscious bias might affect some of the clinical decisions or behaviors of the students as healthcare professionals.*


**Hypothesis** **4.**
*Added attention, in written reflections, to the associations between service-learning projects and basic sciences courses will be accompanied by students’ realization that performing service learning causes them to study for all of their courses with more interest.*


## 2. Methods

### 2.1. Participants

Thirty-four prospective medical students were studied in a required Medical Humanities course from August through December 2019. The students were enrolled in a Master of Science in Biomedical Sciences (MSBS) program at Rocky Vista University in Parker, Colorado, US. The average age of the students was 24.2 years (range 21 to 30 years), 52.9% were female, 70.6% identified as white, 14.7% were Asian, 11.8% were Hispanic, and 2.9% were Black/African American. The humanities course was designed to promote the development of personal and professional skills and identities in teams of students through readings, quizzes, discussions about communication and implicit bias, and written reflections on service to the community.

### 2.2. Team Formation and Procedure

Teams of five or six students were formed for group studying and projects in their Human Anatomy lab and Physiology course, and these teams began meeting for work in Medical Humanities on the first day of that class. Teams selected and performed service-learning projects, and they worked in humanities to take team quizzes in a team-based learning format [17,18].

Each team member performed at least five hours of service alone, with one or more teammate(s), or with their entire team. Teams met initially to select and plan their projects and then regularly at 4- to 5-week intervals to discuss the progress of their projects and team members’ experiences with the projects. Teams scheduled these meetings at their convenience, and the sessions were used to discuss the students’ written reports and critical reflections concerning planning and performing their community service.

Teams generated written minutes and reflections from each meeting. In these observations, teams also integrated their service-learning-related experiences with the content of courses in the MSBS program. For example, students were encouraged to keep an open mind in their Molecular Basis of Medicine course, so many of them tried not to make premature judgements about the people they were serving. To allow students to decide how best to reflect on their experiences, only this expectation of integration with other courses and our definition of self-examination below were provided to them and their teams.

For every two hours of community service, each student produced more than one page of reflections, and each team meeting led to at least one page of written team observations. Fifty-two percent of students’ grades in the one credit-hour Humanities course were based on their written minutes and reflections. Teams sent the course director minutes and individual and team reflections at the ends of the 2nd, 6th, 11th, and 16th week of the 17-week semester.

### 2.3. Qualitative Assessment of Dissonance and Critical Reflection

As described previously [8], minutes and reflections were examined for the presence of dissonance, self-examination, critical reflection, dissonance reconciliation, and bias toward other people or venues [10,19,20]. According to our definition, self-examination (including resultant compassionate behavior) was exhibited when a student recognized how their thoughts (and actions) did not match their personal and humanistic values, experienced perplexity, doubt, hesitation, or mental difficulties (i.e., dissonance), and began to decide how better to align their thoughts (and behaviors) with their values (i.e., dissonance reconciliation) [7,9,10]. That is, both dissonance and dissonance reconciliation had to be present for self-examination (and compassionate behavior) to be present. This definition is consistent with the model of reflection proposed by Nguyen and associates [21].

Grades ranged from 80% (thinking and reflection but no self-examination exhibited) to 100% (self-examination shown). Partial self-examination (e.g., dissonance only) earned a score of about 90%. The values assigned by independent assessors for self-examination correlate well, as we reported elsewhere (*r* = 0.92, [20]).

### 2.4. Quantitative Measurement of Reflective Capacity (RC), Empathy, and Attitudes Toward Service to the Community, Team-Based Learning, and Unconscious Bias

To determine whether reflection on service learning correlated with an increase in students’ RC, we provided students opportunities to complete the 40-item Reflective Practice Questionnaire (RPQ) [11,12] at the beginning of the Medical Humanities course in August 2019 and after completion of the course in December 2019. Along with RC, the RPQ reliably measures the desire for improvement, general confidence, confidence communicating with patients/clients, uncertainty, stress interacting with patients/clients, and job satisfaction (Cronbach’s alpha reliability values range from 0.75 to 0.91 for the subscales among the general population and among a more homogeneous sample of medical students [11,12]; see Rogers et al. [12] online for the current version of the RPQ).

Students also completed the Jefferson Scale of Empathy (JSE, HPS-Version), a valid and reliable measure of cognitive empathy [22], at the beginning and end of the Medical Humanities course (Cronbach’s alpha reliability value of about 0.84 for numerous samples [22] and specifically for a sample of pharmacy students [23]; see Fjortoft et al. [23] online for the current version of the survey). The JSE and the RPQ were marked by students with an ID code in August and were stapled together in December so each student’s RPQ scores could be matched to their JSE scores.

Finally, students completed an in-house survey of their attitudes toward team service-learning projects and unconscious bias (Table 1) after they finished the course in December 2019. This survey was also stapled to the other surveys so responses could be matched with RPQ and JSE scores. Completion of the surveys by students was optional and anonymous. The response rates were 100% for all three surveys, although one student’s numerical responses were not recorded properly and could not be included in assessment of the data.

### 2.5. Statistical Analysis

Statistical analyses were performed using GraphPad Prism 8.0.2 Software, Inc. (La Jolla, CA). Whether or not the students’ median survey opinions—about their team service-learning projects and implicit bias—differed significantly from neutral were determined using one-sample Wilcoxon tests, and the medians of these opinions were compared statistically using the Kruskal–Wallis statistic. Similarly, we determined the median frequencies at which students expressed dissonance, self-examination, dissonance reconciliation, bias mitigation, and compassionate behavior (out of four opportunities to do so). ROUT (Q = 1%) was used to determine whether the students’ values were statistically significant outliers in each set of data.

The students’ initial mean RC and JSE scores in August were compared to their final mean scores in December using unpaired t-tests. These t values were used to calculate effect size (ES) as an *r* value using the GraphPad Prism 8.0.2 Software [24]. The correlations between and among RC, JSE, and other scores measured by the RPQ and in-house survey items were determined as Pearson *r* values using the same software.

This study was reviewed and found to fulfill the criteria for exemption by the Rocky Vista University Institutional Review Board (IRB). Written informed consent was obtained from 32 of 34 students to analyze, report, and publish their written reflections for the Medical Humanities course. Two students left the MSBS program before the end of the humanities course in December 2019, so their written reflections could not be included in this study.

## 3. Results

Our data clearly support each of our four hypotheses.

Hypothesis 1a as in our prior study [8], students experienced dissonance in virtually all of their opportunities to reflect (i.e., they recognized that their thoughts or behaviors did not match their personal and humanistic values, and they experienced perplexity, doubt, hesitation, or mental difficulties). Each of them then used self-examination to reconcile the differences between their thoughts or behaviors and humanistic values in most or all four of their reflections (data not shown).

Implicit bias recognition was a frequent source of dissonance (e.g., item 10 in Table 1), as was the case in our prior study [8]. The students said they became aware of the biases summarized in Table 2 as a result of their community-service experiences. That is, these implicit biases were made more explicit through the selection and performance of a team service-learning project. The single student who did not agree with item 11 in Table 1, “Unconscious bias might affect some of my clinical decisions or behaviors as a healthcare professional,” also stated, “Now that I am able to be more conscious about it, I hope to catch it sooner.” Hence, students seemed to recognize that, with work, their implicit biases could be mitigated even if they were not eliminated. (See hypothesis 3 below.)

Examples of how students faced issues to help them mitigate their biases included;


*Upon arriving at the venue for the MS Muckfest, the sense of community and enthusiasm of the participants changed my perspective about volunteering as I had mentioned in my previous reflection. However, I now realize that I had biases during that time that I was not even aware of till days after the event. As we were waiting for our volunteer assignments, besides wishing that I would still get to work with my classmates, I also preferred to be with people my age…*


And;


*both on our way to the mission and on our way home, we would always discuss how we felt during our time at Denver Rescue Mission and how it either helped us face our biases or if we came away with any additional thoughts….*
(See appendices for more complete examples of reflections)

Hypothesis 1b in the survey concerning team and service learning, the students expressed highly skewed feelings supporting hypothesis 1b, as was the case in our prior study [8]. All students agreed with the statements, “Having a team service-learning project in Medical Humanities was very engaging” and “Next year, Medical Humanities should continue to expect teams of MSBS students to perform service-learning projects and to write reflections on their experiences with the projects.”, whereas all students disagreed with the statement, “I gained very little from our service-learning project and written reflections on the project” (Items 1, 3, and 5 in Table 1).

Moreover, all but two students disagreed, and 61% disagreed strongly, with the statement, “I would have been better off on another team in Medical Humanities,” and virtually all of them agreed that “All things considered, I could not have been assigned to a stronger team in Medical Humanities” (Items 2 and 4 in Table 1). Consequently, all but one student agreed that “Medical Humanities should continue to use team-based learning in future courses,” and no student disagreed with this statement (Item 6 in Table 1). Interestingly, stronger team orientations were associated with higher RC and empathy scores (*r* > 0.47, *p* < 0.01).

Similarly, 29 of the 31 students agreed with the statements “Encounters with people/venues in our service-learning project helped me to see my potential biases toward people/venues more clearly”, “Writing reflections on our service-learning project fostered my professional development”, and “Encounters with people in our service-learning project will help me to be engaged with people regardless of the setting or disposition of the person”, and no student disagreed with these statements (Items 7, 9, and 10 in Table 1).

Hypothesis 1c consistent with hypothesis 1c [8], RC increased in prospective medical students in association with our Medical Humanities course (Figure 1, *r* = 0.26, *p* = 0.02). Also consistent with hypothesis 1a, the results of this survey support the conclusion that more compassionate behavior occurred in association with reflections on team service-learning. Self-appraisal and reflection-on-action are components of the RC survey [11,12], and both self-appraisal (*r* = 0.20, *p* = 0.05) and reflection-on-action (*r* = 0.28, *p* = 0.013) contributed to the increase in RC observed in prospective medical students.

Reflection-with-others also increased significantly (*r* = 0.27, *p* = 0.016), supporting the notion that interactions with teammates formed an essential dimension of students’ personal and professional development owing to reflection on service learning in our course (Hypothesis 1b above).

Hypothesis 1d as reported previously for these students [8], RC, self-appraisal, and reflection-on-action scores each correlated strongly with JSE scores at the beginning of the Medical Humanities course in August 2019 (*r* = 0.72, 0.72, and 0.69, respectively, *p* < 0.0001). These correlations were maintained in December 2019 (*r* = 0.46, 0.35, and 0.52; respectively; *p* = 0.01, *p* = 0.05, and *p* = 0.002). Similarly, JSE scores correlated significantly with reflection-with-others scores in December 2019 (*r* = 0.41, *p* = 0.02). (See further consideration of JSE scores below.)

Hypothesis 2 the increases in students’ RC scores (Figure 1) were associated with increases in their JSE scores (*r* = 0.20, *p* < 0.05). An issue arose, however, with two items on the JSE that comprise the third of three factors sometimes measured by the JSE [25,26]. This third factor, termed “standing in the patient’s shoes,” employs the reverse-score items 3 “It is difficult for a health care provider to view things from patients’ perspectives” and 6 “Because people are different, it is difficult to see things from patients’ perspectives” [23].

A crucial focus of our Medical Humanities course was on effective communication, and we emphasized the importance of understanding patients’ perspectives. We also stressed, however, that it is especially challenging to see their point of view, and one must work hard to do so. The appreciation of this difficulty was emphasized in many of the students’ written reflections (see appendices for hypothesis 4 below), and our students learned how their implicit biases might act further to impair communication with many categories of patients (hypothesis 3 below).

Similarly, students read four books for our course that also emphasized these difficulties in communication (e.gs., If I Understood You Would I Have This Look on My Face? and What Patients Say, What Doctors Hear) [27,28,29,30]. In this context, it is difficult to decide whether “strongly disagree” or “strongly agree” is the most empathetic response to items 3 and 6. (See italicized details of the items above.) Hence, we analyzed our data without these items and found that the increase in students’ JSE scores in association with our course was closer to moderate practical importance without them (Figure 2 and see above).

Students’ JSE (with or without items 3 and 6) and RC scores also correlated positively with the desire for improvement and job satisfaction, as measured by the RPQ (*p* < 0.05). As we discussed previously, higher cognitive empathy is inversely associated with burnout in healthcare professionals [31].

Hypothesis 3 all but two students agreed with the statement, “Unconscious bias might affect some of my clinical decisions or behaviors as a healthcare professional,” and only one disagreed (Item 11 in Table 1). The student who disagreed commented, “Now that I am able to be more conscious about it, I hope to catch it sooner.” Thus, even that student seemed to recognize that their implicit biases might affect their decisions and behaviors. Moreover, students frequently described discovery of biases they held toward others in their written critical reflections (Table 2 and Appendix A and Appendix B).

Hypothesis 4 response to item 8 in Table 1, “Encounters with people in our service-learning project caused me to study for all of my courses with more interest than likely would have occurred without the project,” the class as a whole agreed and their median answer was significantly higher than neutral (i.e., 4.0; *r* = 0.67; *p* < 0.0001). The students’ responses to item 8 were also more positive in 2019 than in prior years of the humanities course when students were neutral and not overtly expected to see the connections of service-learning to other courses (Mann–Whitney test; 2017 vs. 2019, *r* = 0.35, *p* < 0.01; 2018 vs. 2019, *r* = 0.36, *p* < 0.01). Finally, the students’ increased engagement with other courses in the MSBS curriculum in 2019, in association with their service-learning projects, is exemplified in their written critical reflections concerning Physiology in Appendix C and Appendix D.

## 4. Discussion

Implicit bias remains a nearly intractable problem in the US [14,32,33] and elsewhere [34,35]. Healthcare professionals hold negative attitudes toward people of color and discriminate against them. Such biases adversely affect the treatment of patients and their adherence to treatment plans leading to poorer outcomes of their care. Even in residency selection, implicit biases toward applicants may deprive patients of care by the most qualified healthcare providers [36,37]. Our methods to foster implicit bias mitigation and compassionate behavior by virtually every prospective medical student could, however, also be used to train all healthcare professionals. This pedagogy would improve public health in general and the health of people of color in particular. To do so, however, requires cultural changes in medical education. Efforts to mitigate implicit bias through reflection on behavior, such as service to the community both within and outside of their practices, should continue throughout providers’ careers.

We used implicit association tests as tools to help students mitigate biases not only against people of color, but also toward other groups experiencing discrimination for their gender, body weight, and sexual orientation. Then students expanded the list further—through experiences in their service-learning projects—to include children, older people, homeless people, and those of lower socioeconomic status (Table 2). Thus, the groups of people against whom there is prejudice, and a need to mitigate such bias, expand depending on the circumstances of healthcare delivery. Toward this end, prospective medical students in our course viewed the last hour of a video from the Gates Foundation concerning discrimination owing to implicit biases, and work to mitigate them, in a variety of settings around the world. (See the last hour of the video at the following site: https://youtu.be/1SbUSj5iEgs)

Our methods resemble a pedagogy of discomfort [13,14], and our results mirror the transformative learning model of Sukhera and associates to be published soon [38]. In this model, a disorienting experience, such as service learning and discovery of an unconscious bias, leads to self-examination and critical reflection, as performed by our prospective medical students. The students then worked to acquire new skills and exhibit more compassionate behavior toward the people they chose to serve, as outlined in Figure 1 of [38]. These authors propose the transformative learning theory they discuss as a guide for implementing educational strategies to combat harmful implicit bias in healthcare professions [38].

Prospective medical students were eager to implement their new skills, as exemplified in the subsequent semester of their program. During that semester, students were required to attend a three-hour evening presentation on implicit bias and diversity organized by medical students at our institution. We then offered them an opportunity to earn a small amount of extra credit by submitting a reflection on their reaction to the presentation as part of their immunology course. In a prime theme of those reflections, students aspired not merely to talk about bias and diversity, but to learn what actions they can take to combat biases against patients and the low level of diversity among healthcare professionals.

A bottom-up approach to institutionalizing training to foster critical reflection, bias mitigation, and compassionate behavior throughout healthcare providers’ careers is to include other courses and rotations in these efforts to educate them as students. In our case, we established service learning and bias mitigation experiences for students that were successful in the opinions of faculty, administration, and, perhaps most importantly, students. We then worked to have other faculty members begin to accept the idea that such experiences could become part of their courses. So far, this plan has included use of the same groups/teams to study and perform course projects together, such as in Anatomy and Physiology, and to relate service-learning experiences, including dissonance, bias mitigation, and compassionate behavior, to activities in those other courses (e.g., Appendix C and Appendix D).

Eventually, however, we expect other faculty members to adopt a plan to incorporate service learning into their courses and to make written critical reflections on the experiences an aspect of student development they assess. At our institution, this plan has already been applied in the Immunology course for prospective medical students. Another selling point will be emerging data in our courses showing that service learning and related experiences cause students to study material in all of their courses with more interest (Item 8 in Table 1). It is well established that courses, employing critical reflections on service-learning experiences by students, enhance their academic performance as well as their non-cognitive development [39,40,41,42]. Since most of the same faculty members direct courses in other programs at our institution, a mechanism exists to institutionalize our approaches for all of our healthcare professional trainees at least for the basic sciences.

But how would such a program be continued in clinical sciences? At our institution, students can be very impactful on the curriculum and the educational environment, especially when it comes to the quality of their rotations. We believe that a robust program of critical reflection on service-learning and bias mitigation in basic sciences would lead students to expect such a plan to continue into their clinical years. They value their learning teams so much that they would likely want to continue these written and verbal learning experiences with support from teammates they had in basic sciences (recall Items 2, 4, and 6 in Table 1 and see the student’s reflection in Appendix E). Such sharing of stories among healthcare professional students [27] has been used in other contexts to foster their interdisciplinary collaboration [43].

## 5. Limitations

Our results may seem, at first, challenging to generalize to other pre-professional and professional healthcare programs at other universities. We studied a total of only about 90 prospective medical students in a Medical Humanities course at a single university. Our results for the first two cohorts of students [8] were, however, replicated in the present study. The completely reproducible nature of our data makes it likely that we would obtain similar results for other programs at our institution.

Moreover, the ability of Biochemistry team service-learning experiences to foster self-examination and compassionate behavior in medical, pharmacy, and prospective medical and dental students at another university [6,7], with samples of students exceeding 200, supports the theory that our approach could be implemented more broadly. We encourage others to test the following hypotheses further: teams of pre-professional and professional healthcare students (1) build trust and psychological safety from shared provocative experiences and (2) display frequent self-examination and compassionate behavior when they are free to make meaning of the experiences for themselves.

## 6. Conclusions

The experiences of selecting and performing team service-learning projects produced dissonance among virtually all prospective medical students. This dissonance arose, in part, from team and class discussions of provocative readings involving biases and the results of implicit association tests. Dissonance led to self-examination and compassionate behavior to reconcile the conflict in virtually all students. Furthermore, students’ reflective capacity scores grew in association with our Medical Humanities course, and they were accompanied by increases in students’ cognitive empathy scores (a component of compassion). Team and class discussions of the difficulties in mitigating unconscious biases, and the effect of these biases on patient–provider communication, also led virtually all students to accept that implicit biases might influence some of their clinical decisions and behaviors as healthcare professionals. Finally, added attention, in written reflections, to the associations between service-learning projects and basic sciences courses helped students realize that performing service learning caused them to study for all of their courses with more interest. Students also clearly benefited from the psychological safety, support, and trust of each other that they built in teams to study and complete projects in several basic sciences courses as well as in Humanities.

## Figures and Tables

**Figure 1 ijerph-17-02169-f001:**
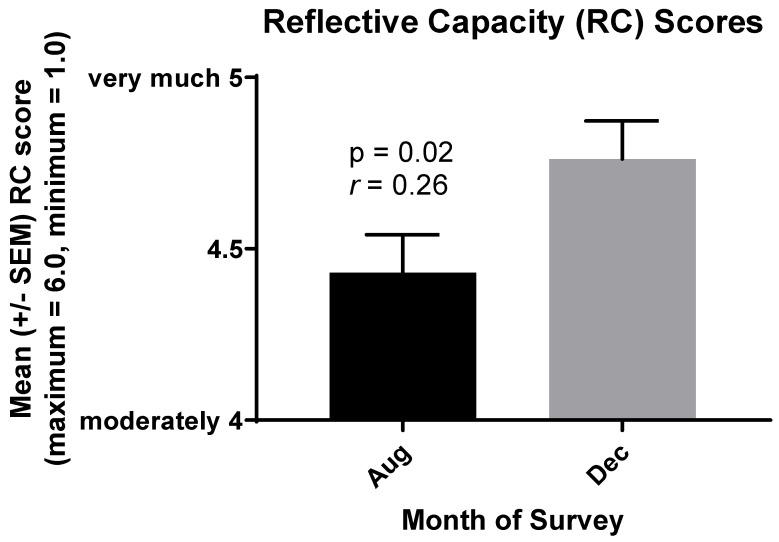
Increase in reflective capacity scores of prospective medical (MSBS) students in association with written reflections on service learning between August and December 2019. The mean response of MSBS students increased from closer to “moderately” in August to nearer “very much” in December for 16 statements concerning self-reported reflection.

**Figure 2 ijerph-17-02169-f002:**
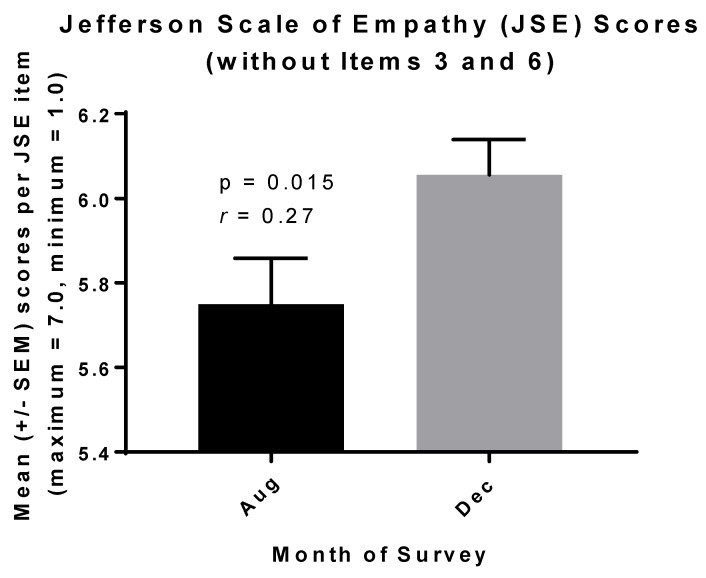
An increase in the cognitive empathy (JSE) scores of prospective medical (MSBS) students occurred in association with the rise in their reflective capacity (RC) scores (Figure 1) between August and December 2019.

**Table 1 ijerph-17-02169-t001:** Medians (bold italics) and distributions of responses to statements regarding team and service learning were each significantly different from neutral (i.e., 4.0) for all 11 items (*n* = 31, *p* < 0.0001). * statistically significant outliers.

Strongly Disagree	Disagree	Somewhat Disagree	Neither Agree/ Disagree	Somewhat Agree	Agree	Strongly Agree	
1.	Having a team service-learning project in Medical Humanities was very engaging.						
0	0	0	0	1 (3.2%) *	8 (25.8%)	22 (71.0%)	
							**7.0**
2.	I would have been better off on another team in Medical Humanities.						
19 (61.3%)	7 (22.6%)	3 (9.7%)	1 (3.2%)	1 (3.2%) *	0	0	**1.0**
3.	Next year, Medical Humanities should continue to expect teams of MSBS students to perform service-learning projects and to write reflections on their experiences with the projects.						
0	0	0	0	1 (3.2%) *	6 (19.4%) *	24 (77.4%)	**7.0**
4.	All things considered, I could not have been assigned to a stronger team in Medical Humanities.						
0	0	0	2 (6.5%) *	3 (9.7%) *	5 (16.1%)	21 (67.7%)	**7.0**
5.	I gained very little from our service-learning project and written reflections on the project.						
21 (67.7%)	8 (25.8%)	2 (6.5%)	0	0	0	0	**1.0**
6.	Medical Humanities should continue to use team-based learning in future courses.						
0	0	0	1 (3.2%) *	1 (3.2%) *	4 (12.9%) *	25 (80.6%)	**7.0**
7.	Writing reflections on our service-learning project fostered my professional development						
0	0	0	1 (3.2)	4 (12.9%)	12 (38.7%)	14 (45.2%)	**6.0**
8.	Encounters with people in our service-learning project caused me to study for all of my courses with more interest than likely would have occurred without the project (*n* = 30).						
0	2 (6.7%)	2 (6.7%)	4 (13.3%)	6 (20.0%)	7 (23.3%)	9 (30.0%)	**6.0**
9.	Encounters with people in our service-learning project will help me to be engaged with people regardless of the setting or disposition of the person.						
0	0	0	2 (6.5%) *	1 (3.2%)	13 (41.9%)	15 (48.4%)	**7.0**
10.	Encounters with people/venues in our service-learning project helped me to see my potential biases toward people/venues more clearly.						
0	0	0	2 (6.5%)	4 (12.9%)	8 (25.8%)	17 (54.8%)	**7.0**
11.	Unconscious bias might affect some of my clinical decisions or behaviors as a healthcare professional.						
0	1 (3.2%) *	0	1 (3.2%)	2 (6.5%)	15 (48.4%)	12 (38.7%)	**6.0**

**Table 2 ijerph-17-02169-t002:** Summary of written statements of prejudices expressed by students in a survey regarding the biases of which they became aware in their team service-learning experiences (30 of 32 students stated one or more of their biases).

Nature of Negative Bias	Number of Times Expressed
Age (children and older adults)	10
Homeless People	7
Culture/Race	6
Socioeconomic status	4
Obesity	3
Gender	3
Hygiene	1
Mental health patients	1
Men I do not know	1
Environmental	1

Other Statements; Lots; The bias of my expectation going into an experience; That it is impossible to avoid bias; My bias to stay with people I already know.

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
