# Peer review of "Fostering Empathy, Implicit Bias Mitigation, and Compassionate Behavior in a Medical Humanities Course"

_ijerph, 2020, doi:10.3390/ijerph17072169_

Round 1

Reviewer 1 Report

This is a very interesting, thought provoking paper and it addresses an important topic. However, it has the flaws that research using self-report data with small samples always has. The research was conducted in a course with a group of students enrolled in a (required?) course and it is a relatively small sample. Also, the data are based on "service learning" experiences. Such experiences often range from very good to very bad and usually the instructor does not have much control over the quality of those experiences. Thus there is no objective way to learn whether all students had similar experiences in the service learning sites. Also, self-report data is usually difficult to measure objectively. The paper is well written, however, and its limitations are well described. There also is a good discussion of the problem of bias in professionals in this field. The paper would be strengthened if there were a set of self-report examples included since it would give the reader a better sense of the issues faced that may contribute to bias and loss of bias. 

Author Response

Reviewer 1 (Lines indicated in our italicized responses below are when showing no markup in track changes.)

This is a very interesting, thought provoking paper and it addresses an important topic. However, it has the flaws that research using self-report data with small samples always has. The research was conducted in a course with a group of students enrolled in a (required?) course and it is a relatively small sample.

We now state that the course was required in line 92.  The sample size in the present study is 34 students, but our findings replicate and extend our findings with 52 students in prior years of the course [reference 8].  Moreover, we now mention under “Limitations” that team service-learning experiences to foster self-examination and compassionate behavior at another university involved samples exceeding 200 of our medical and 200 of our pharmacy students (line 404).

Also, the data are based on "service learning" experiences. Such experiences often range from very good to very bad and usually the instructor does not have much control over the quality of those experiences. Thus there is no objective way to learn whether all students had similar experiences in the service learning sites.

We would argue that the important similarity in service-learning experiences is dissonance associated with selecting and performing the service.  Every student wrote critical reflections to reconcile this dissonance.  Virtually all students also agreed that their team service-learning was very engaging and helped them see their biases against people.

Also, self-report data is usually difficult to measure objectively. The paper is well written, however, and its limitations are well described. There also is a good discussion of the problem of bias in professionals in this field. The paper would be strengthened if there were a set of self-report examples included since it would give the reader a better sense of the issues faced that may contribute to bias and loss of bias. 

We now include short excerpts from the appendices in the text (lines 206-214) to illustrate how students faced issues to help them mitigate their biases.

Reviewer 2 Report

It is an interesting topic. The following are recommendations to improve the article. Thank you.

ABSTRACT
In relation to the abstract, it should implicitly have the following structure: introduction-method (questionnaires, participants and design, at least) - results - conclusions. The exposed article does not follow this structure, so when reading it, you cannot extract all the potential. The abstract should show the number of participants, sex (in percentage), the instruments used, etc.
With respect to the keywords, they are too many. It would have to be simplified. Similarly, it would be advisable that they be put in alphabetical order and that one of the keywords be the study design.

INTRODUCTION
While it is interesting, it is necessary to introduce more quotes and greater depth. The theoretical framework needs to be expanded. The specific objectives are not clear. The hypotheses are not fully developed or their development is disproportionate: hypothesis one tries to carry out a much larger check than the rest. Perhaps hypothesis 1 could / should be divided into several hypotheses.

METHOD
The participants section is too superficial. The characteristics of the participants need to be further developed. For example, average age, percentage of women and men, marital status ...
In addition, the number used to conduct the study is small. The sample of participants should be increased to achieve greater external validity. In fact, the small number even harms the use of certain tests whose requirement is based, among other parameters, on a minimum number of participants, for example, parametric tests.

In section 2.4, the internal consistency of the instruments used should appear.

RESULTS

You have to review the citation regulations and the format of some words. For example, Table 1.
Line 162 is incomplete.

Check the location / size of the sentences of lines 209-212

DISCUSSION AND CONCLUSIONS

Line 323. Review the size and format supported by the magazine.

Analyze if the appendices provide relevant information for the study. Perhaps it would be a good idea to perform a more quantitative analysis of the information presented in them, for example, through the Atlas program

Author Contributions statement:

Lines 508-513. In the authors' contributions, the authors should be represented by their initials, not by their full name.

Line 240 is incomplete.

REFERENCES
Less than 50% of references correspond to articles from the last five years. An extension is recommended, which can be used to improve the introduction and in turn the discussion.

Thank you very much for your interest and attention.

Author Response

Reviewer 2 (Lines indicated in our italicized responses below are when showing no markup in track changes.)

It is an interesting topic. The following are recommendations to improve the article. Thank you.

ABSTRACT
In relation to the abstract, it should implicitly have the following structure: introduction-method (questionnaires, participants and design, at least) - results - conclusions. The exposed article does not follow this structure, so when reading it, you cannot extract all the potential. The abstract should show the number of participants, sex (in percentage), the instruments used, etc.

The abstract has been rewritten to include these elements.

With respect to the keywords, they are too many. It would have to be simplified. Similarly, it would be advisable that they be put in alphabetical order and that one of the keywords be the study design.

The number of keywords has been decreased from 8 to 6 and one is the study design.

INTRODUCTION
While it is interesting, it is necessary to introduce more quotes and greater depth. The theoretical framework needs to be expanded.

We expanded our theoretical framework to include the important concept of the “pedagogy of discomfort” which can be generated by considering unconscious bias (lines 55-56).

The specific objectives are not clear. The hypotheses are not fully developed or their development is disproportionate: hypothesis one tries to carry out a much larger check than the rest. Perhaps hypothesis 1 could / should be divided into several hypotheses.

We would like to keep the structure of our hypotheses since one of our purposes was to determine whether hypotheses, supported by data in our prior study, would be confirmed in the present study.  We attempted to clarify this point in sections stating our hypotheses including lines 69-70.

METHOD
The participants section is too superficial. The characteristics of the participants need to be further developed. For example, average age, percentage of women and men, marital status ...

We now include demographics of students in lines 94-96.

In addition, the number used to conduct the study is small. The sample of participants should be increased to achieve greater external validity.

The sample size in the present study is 34 students, but our findings replicate and extend our findings with 52 students in prior years of the course [reference 8].  Moreover, we now mention under “Limitations” that team service-learning experiences to foster self-examination and compassionate behavior at another university involved samples exceeding 200 of our medical and 200 of our pharmacy students (line 404).

In fact, the small number even harms the use of certain tests whose requirement is based, among other parameters, on a minimum number of participants, for example, parametric tests.

Larger samples are indeed needed to show surveys to be reliable, but this purpose was not among our hypotheses.  Sample sizes in excess of 30 are sufficient for t-tests.

In section 2.4, the internal consistency of the instruments used should appear.

Cronbach’s alpha reliability values are now reported for the RPQ and JSE, HPS-Version in lines 145-147 and 150-151.  The in-house survey was designed to collect student opinions and was not designed to measure a single characteristic such as reflective capacity or empathy.

RESULTS

You have to review the citation regulations and the format of some words. For example, Table 1.
Line 162 is incomplete.

Check the location / size of the sentences of lines 209-212

The layout of the manuscript was altered on conversion from our format to that of the journal.  We have corrected these alterations as shown in track changes. (Please view the manuscript also without showing track changes markup.)

DISCUSSION AND CONCLUSIONS

Line 323. Review the size and format supported by the magazine.

The font and font size have been corrected in line 349.

Analyze if the appendices provide relevant information for the study. Perhaps it would be a good idea to perform a more quantitative analysis of the information presented in them, for example, through the Atlas program

Complete analyses of students’ written reflections were reported in our prior work [reference 8].  The results were similar in the present study and are not shown again, although we refer to them in lines 192-196.

Author Contributions statement:

Lines 508-513. In the authors' contributions, the authors should be represented by their initials, not by their full name.

When submitting our manuscript, we were instructed to copy this section from where it was created by the journal software, and paste it into our manuscript.  We have changed our full names to our initials in this section (lines 537-539).

Line 240 is incomplete.

Again, the layout of the manuscript was altered on conversion from our format to that of the journal.  We have corrected these alterations as shown in track changes.  Lines 261 and 267 show the correction for the line mentioned above when viewing no markup in track changes.

REFERENCES
Less than 50% of references correspond to articles from the last five years. An extension is recommended, which can be used to improve the introduction and in turn the discussion.

As stated above, we expanded our theoretical framework in the introduction to include the important concept of the “pedagogy of discomfort” which can be generated by considering unconscious bias (lines 55-56).  This concept is supported by two recent references in the introduction and in the discussion, line 350. 

In addition, we added a recent reference [43], at the end of our discussion, which emphasizes the importance of sharing stories among healthcare professional students (lines 392-394). 

Thank you very much for your interest and attention.

Thank you for your comments and suggestions.

Round 2

Reviewer 2 Report

Thank you very much for your interest and effort.